# Parental bonding in retrospect and adult attachment style: A comparative study between Spanish, Italian and Japanese cultures

Maria Alejandra Koeneke Hoenicka[1ʘ], Oscar López-de-la-Nieta[1,2ʘ], José Luis Martínez Rubio[1], Kazuyuki Shinohara[3], Michelle Jin Yee Neoh[4], Dagmara Dimitriou[5], Gianluca Esposito[6], Giuseppe Iandolo[1,7] *

1 Department of Psychology, School of Biomedical Sciences, European University of Madrid, Madrid, Spain, 2 Clinical Research & Diagnosis Division, SerenaMente Psychology & Consulting Service, Pinto, Madrid, Spain, 3 Department of Neurobiology & Behavior, School of Biomedical Sciences, Nagasaki University, Nagasaki, Japan, 4 Psychology Program, School of Social Sciences, Nanyang Technological University, Singapore, Singapore, 5 Sleep Education and Research Laboratory, University College London Institute of Education, London, United Kingdom, 6 Affiliative Behaviour and Physiology Lab, Department of Psychology and Cognitive Science, University of Trento, Rovereto, Italy, 7 Observation and Functional Diagnosis Division, PSISE Clinical and Developmental Psychological Service, Madrid, Spain

ʘ These authors contributed equally to this work.
* giuseppe.iandolo@universidadeuropea.es

**Data Availability Statement:** The dataset generated during the current study is available at

## Abstract

Attachment is an innate human relational mechanism that develops progressively from early childhood, influences individuals' representations and behaviors, shapes relationships, and affects the social and cultural environment. Parental bonding refers to the ability of parents to be emotionally and behaviorally available to the child during infancy. Attachment style refers to the individual's relational attitude in close relationships that influences adult love, bonding, handling relationships, and social exploration. The role of intergenerational, cultural and developmental factors influencing the relationship between the attachment style in adulthood and the parental bonding style recalled during childhood has been debated. This study explores the relationships between recalled parental bonding, adult attachment style, and cultural background in a sample of Spanish, Italian, and Japanese adults using a cross-sectional and cross-cultural design. For this purpose, the validated versions of the Experience in Close Relationship Scale and the Parental Bonding Instrument were administered to a non-clinical population of three hundred and five participants in the three countries. Results show that the most frequent adult attachment style is the secure style, followed by the dismissing-avoidant, the preoccupied, and the fearful-avoidant style. The dismissing-avoidant style was the most frequent insecure attachment style in the Japanese sample whereas the preoccupied style was the most frequent insecure attachment style in the Italians and Spaniards. Japanese are more anchored to the memory of maternal and paternal overprotection, which is related to more avoidance in actual close relationships. Spaniard's current relationships are mildly independent of recalled parental bonding, showing an association between lower current avoidance to primary parental care. In the Italian sample,

the link: https://figshare.com/s/
3c1c5782dc9218893246.

**Funding:** The author(s) received no specific
funding for this work.

**Competing interests:** The authors have declared
that no competing interests exist.

there is no significant relationship between current adult close relationships and recalled
parental bonding. These results suggest that different cultural models influence adult attach-
ment representations differently, in terms of the weight placed on attachment-related avoid-
ance, attachment-related anxiety, care, and overprotection in infant and adult relationships.

## Introduction

Attachment theory is one of the "grand theories" of social and personality development that
has endured the passage of time [1–7]. It attracts research attention from social, personality,
and developmental psychology, giving rise to different methodological traditions in the study
of infant and adult attachment [3, 8–16].

Although recalled parental bonding and attachment style are sometimes wrongly used
interchangeably, there are essential differences between them [17]. Parental bonding refers to
the ability of parents to be available, emotionally and behaviorally, to the infant's demands and
needs of care, protection, and exploration [18–21]. Early experiences with parents give rise to
different attachment blueprints that supply an implicit relationship pattern in adulthood,
which are also referred to as Internal Working Models (IWM) [16, 22, 23].

Based on how skillful and emotionally available parents are in attending to their child's
demands, considering the child's responsiveness and implication, each child forms a mental
representation with respect to two dimensions: oneself and others [10, 24, 25]. The first dimen-
sion is whether they deserve (worthy) affection, care, and attention; the second is whether oth-
ers are capable, willing, and available to give that affection, care, and attention [16, 26, 27]. The
perception of directionality of love with early attachment figures [23, 28] shapes an individual's
IWM [29], influencing adult attachment through updates and revisions over the lifespan [1,
30, 31].

Several studies analyzed the relationship between attachment style in adulthood and
recalled parental bonding during childhood [20, 32–35]. Many studies underline the relation-
ship between early attachment patterns, retrospective parental bonding representations, and
attachment style during adulthood [2, 33]. The scientific literature confirms a correlation
between the early child behaviors during the Strange Situation with caregiver [7] and the struc-
ture of the mental representations during the lifespan, as well as an intergenerational and cul-
tural correlation with the parents' attitude [2, 36, 37]. Secure-autonomous styles develop as a
result of appropriate early parenting, while insecure styles derive from less relevant early
attachment experiences, indicating a relationship between recalled parental bonding and
attachment in adulthood [38–45].

These models arise from the original Bowlby and Ainsworth's classification of secure, inse-
cure-avoidant, insecure-ambivalent, and insecure-disorganized attachment styles [2, 3, 7, 29,
46, 47]. Based on the relevance of mental representations during the life cycle, [27] classified
attachment according to the view of the self and others, clustering adult attachment style into
four types: secure, insecure-dismissing, insecure-preoccupied, and insecure-fearful-avoidant.
The four patterns of Bartholomew and Horowitz's model [27] are an evolution of Hazan and
Shaver's three-factor perspective [15]: the secure attachment, the anxious-ambivalent, and pre-
occupied patterns are analogous in both models. In Bartholomew and Horowitz's model,
Hazan and Shaver's anxious-ambivalent pattern is later split into the dismissing and fearful
patterns [1, 48].

Subsequently, [26] translated these attachment patterns into a two-dimensional model of attachment-related behaviors. Their two-factor model consisted of attachment-related anxiety (the degree to which people are insecure about a partner's availability, love, and responsiveness) and attachment-related avoidance (the strategies people use to regulate attachment-related behavior, thinking, and affect). Regarding this last dimension, it is essential to note that avoidant people tend to display behaviors such as anger or withdrawal to avoid conflict in the couple relationship [49].

According to Brennan, Clark, and Shaver's model, secure individuals show a positive view of self and others (low attachment-related anxiety and low attachment-related avoidance). Dismissing-avoidant individuals show a positive view of self and a negative view of others (low attachment-related anxiety and high attachment-related avoidance). Preoccupied people show a negative view of self and a positive view of others (high attachment-related anxiety and low attachment-related avoidance). Fearful-avoidant persons show a negative view of self and others (high attachment-related anxiety and high attachment-related avoidance). Attachment representations can change as a function of life experiences, but the person can still display specific vulnerabilities, behaviors, and implicit relational patterns in adulthood, related to IWM that developed during the early attachment experience [30, 50–54].

Longitudinal studies indicate that early family interactions are predictors of adult attachment and romantic behavior [33, 34, 55]. The memory of parental care (affection, affective support, empathy, and closeness) and control (overprotection, intrusiveness, excessive contact, infantilization, hindering of autonomous behavior) affects adulthood [56–58]. There are gender differences in the effect of recalled parental bonding on adulthood [59], and it is related to the development of certain psychopathologies [38, 39, 42–44]. Between genders, there are no differences between secure and insecure attachment [15, 27]. However, in the context of insecure attachment, men tend to be more dismissing-avoidant, and women are more anxious-preoccupied across cultures [60, 61].

Across samples differing in age and geographic region [62], findings on the relationship between recalled parental bonding and adult attachment have been inconsistent. Specifically, concerning the present study, previous research found an association between Bowlby's IWM and Bartholomew and Horowitz's model of adult attachment [32, 63]. Although some authors argue that there should be a relationship between IWM and adult attachment styles, other empirical data do not support this idea [55, 64]. Findings that do not support this idea may be the result of lifelong learning and changes to attachment styles over the lifespan since IWM refer to the internalized mental representations based on initial attachment figures, that individuals have about the self and others. However, specific vulnerabilities to specific stimuli still activate the IWM as an automatic response to contexts in which the individual interprets a danger to the relationship, modifying the adult attachment style temporarily in favor of the original IWM [52].

In conclusion, the relationship between infant attachment behavior and adult representations of close relationships remains a highly debated topic [32, 65]. This relationship could possibly be mediated by different intermediate variables like life experiences, intergenerational and cultural factors [2, 36, 37], contributing to the inconsistent findings in the existing literature.

## Adult attachment style in different cultures

Culture shapes attachment styles throughout life and in close adult relationships because each culture involves different behaviors, habits and strategies as social patterns in adult life that are considered acceptable for a given human social group [66]. Granqvist [37] argues that

attachment relationships (particularly secure ones) provide an ideal context for cultural transmission and social learning. Intergenerational cultural transmission and social learning are related to the attachment style to satisfy the needs for emotional closeness, a secure base, and a safe haven. Different cultures and social contexts have different notions of the acceptable ways in which relationships are established with an attachment figure [48, 67–74]. For example, in most East Asian countries, the prevailing cultural model is collectivism [75], in which sacrifice for the well-being of the community prevails over benefit for the individual [76]. This cultural model usually goes hand in hand with a family model defined by interdependence, based on a relationship of subordination for the life of children to parents, where implicit loyalty and the commitment to fidelity are critical factors in family stability [48, 77]. According to Western attachment theorists, the Japanese mother-child relationship is considered more dependent than Western cultures [72], and this is due to a more distant quality of the marital relationship, the alteration of which directly affects the bond established with the child [78, 79]. On the other hand, Western Mediterranean cultures such as Italian and Spanish also seem to show a common collectivist perspective on parenting but are more focused on socialization and external interdependence [80–82]. The Italians tend to focus on interactive and affective social exchanges related to the enjoyment of life [83–86], while the Spanish tend to focus on interactive social exchanges related to being a good citizen and family member [81, 83].

Therefore, several studies show differences between Western and Eastern civilizations in the type of relationships established between parents and children and between their subsequent close relationship between adults [70–72]. Some studies found no differences in adult attachment distribution across cultures, whereas others showed cultural variability [87–90]. Ainsworth, Blehar, Waters, and Wall [91] underlined a frequency of 60% of secure attachments in the North American infants evaluated, 20% of insecure-avoidant, and 20% of insecure-ambivalent. Previously, Ainsworth [92] observed that separation anxiety appeared earlier in Ugandan mother-child dyads than in American children due to a closer attachment in the first years of life in Uganda. Research about adult attachment show similar proportions of attachment patterns in the adult population across cultures, depending on the self-report instrument applied, with a range between 48–68% of secure attachment, 11–22% of dismissing, 8–15% of preoccupied, and 13–28% of fearful-avoidant [70, 74, 87, 93, 94].

More extensive studies claim that in the non-clinical population, the most prevalent adult attachment style across cultures is the secure one, around 60% [88–90]. The remaining 40% of the general population shows different tendencies in insecure attachment style, depending on cultural variations. In Europe, dismissing-avoidant attachment is more prevalent [95], observed more in Eastern Europe than in Western Europe. A high prevalence of preoccupied attachment style also predominates in Eastern Europe [96, 97], being most prevalent in Mediterranean cultures [48, 83]. In Northern Europe, preoccupied attachment is also highly prevalent [98]. This same alternation of insecure style is also representative of other continents. In North America, insecure attachment styles range between dismissing-avoidant and preoccupied, depending on the sample [15, 99, 100]. In Asia, insecure attachment is more prevalent in East Asian countries [89, 90], especially the preoccupied attachment style [90, 94]; while dismissing-avoidant attachment is more prevalent in Southeast Asia, as well as in Africa [88, 90, 101]. For example, in Japan, the most prevalent adult attachment style is secure, around 68% [87, 90].

Different studies have highlighted how the Japanese culture is more prone to self-criticism while Western cultures are more prone to self-improvement [66]. Moreover, the relationship between mothers and children in Japan is more dependent than in Western cultures [72, 78, 79]. Furthermore, a disengaged parental couple is socially accepted in Japan [102]. On the other hand, in Spain and Italy, romanticism is culturally considered necessary after marriage and child-rearing [103–105].

Japanese culture generally accepts a more distant partner relationship and a greater mother-child dependency, leading to greater child overprotection. Spanish and Italian cultures generally accept a closer partner relationship and more outstanding mother-child care but not overprotection, considered a form of spoiling [106, 107].

In Japanese culture, this greater child-parental interdependence and parental overprotection translate into an attitude that consists not only in perceiving and responding to the demands and needs of the child but also in anticipating his/her needs and calls for help [69, 72]. In addition, this child-parental interdependence often goes hand in hand with a less romantic and conflictive parental couple [108]. From this family configuration, culturally accepted in Japan, comes a more avoidant than anxious attachment tendency. In this close relationship arrangement, the effort anticipating the other person's needs can involve a persecutor-distancer circle that tends to avoid closeness [27, 59, 108–111].

On the other hand, Spanish and Italian cultures consider parental sensitivity more focused on care, perceiving and responding to the child's signals and requests for help, without anticipating his/her needs [18, 91, 106, 107]. In addition, Spanish and Italian childcare goes hand in hand with a view of the parental couple in a romantic model expected to be enduring after marriage and child-rearing [103–105]. From this family configuration, culturally accepted in Spain and Italy, comes a more anxious than avoidant attachment tendency. In this close relationship structure, the effort to be more concerned about other's well-being implies preoccupations of receiving and giving care to others, without anticipating their needs, but waiting for the social signals that indicate it [80–82, 112, 113].

## Present study

This study explores the relationships among recalled parental bonding, adult attachment style, and cultural background in a sample of Spanish, Italian, and Japanese adults using a cross-sectional and cross-cultural design. The research design explores three hypotheses:

H1. Independent of culture, the secure attachment style is most frequent.

H2. Regarding insecure attachment styles, the most prevalent attachment style in Mediterranean cultures (Italian & Spanish) is the preoccupied style, while in the Japanese culture is the avoidant one.

H3. Attachment-related avoidance in Japanese culture is correlated with higher levels of recalled parental overprotection. In Mediterranean cultures (Italian & Spanish), higher attachment-related anxiety is correlated with higher levels of recalled parental care.

## Methodology

### Participants

Participants were recruited from a voluntary non-clinical sample of adults screened in 2017 (Table 1). All participants completed a brief sociodemographic and clinical screening questionnaire to ascertain participants' educational level and parental bonding experience. All participants who reported previous psychological or psychiatric diagnoses were removed from the sample.

The sample consisted of a total of 305 participants among university students from the European University of Madrid—Spain (32.78%, $N = 100$), the University of Trento—Italy (27.87%, $N = 85$), and the University of Nagasaki—Japan (39.34%, $N = 120$). Of the total sample, 52.05% ($N = 160$) were male, and 47.05% ($N = 145$) were female; the mean age of the entire sample was 22.72 years (SD = 5.28). Descriptive statistics of the sample are summarized in Table 1.

**Table 1. Descriptive statistics of the participant sample.**

| | Age | | | | Sex | | Total |
|---|---|---|---|---|---|---|---|
| | **Min** | **Max** | **Average** | **SD** | **Female** | **Male** | |
| Spain | 17 | 54 | 24.07 | 7.88 | 50 (50%) | 50 (50%) | 100 |
| Italy | 18 | 34 | 22.76 | 3.17 | 59 (69%) | 26 (31%) | 85 |
| Japan | 19 | 37 | 21.55 | 3.08 | 36 (30%) | 84 (70%) | 120 |
| Total | 17 | 54 | 22.72 | 5.28 | 145 (47.5%) | 160 (52.5%) | 305 |

Each participant signed an informed consent form at each data collection site. The study protocol was approved (CEIm PY:17/20) by the Ethics Committee of the Hospital de Getafe (Madrid, Spain).

## Measurements

The study's instruments consisted of three measures: a brief sociodemographic and clinical-screening questionnaire, the Experience in Close Relationship Scale–ECR [26], and the Parental Bonding Instrument—PBI questionnaires [58].

**Experience in Close Relationships Scale.** The Experience in Close Relationships Scale–ECR [26] consists of 36 items on a Likert scale of 7 points ranging from (1) "Strongly disagree" to (7) "Totally agree" and assesses the behaviors of affective exploration in close relationships. The versions of the ECR scale used for this study were ECR-S [114] for Spanish participants, ECR-I [115] for the Italian group, and ECR-J [116] for Japanese participants. The ECR-S Spanish (Cronbach's α Anxiety: 0.85; Cronbach's α Avoidance: 0.87) [114], ECR-I Italian (Cronbach's α Anxiety: 0.89; Cronbach's α Avoidance 0.89) [117], and the ECR-J Japanese (Cronbach's α Anxiety: 0.87; Cronbach's α Avoidance 0.91) [116] show good internal consistency. The ECR-S showed the intended two-factor structure in the study conducted by Alonso-Arbiol et al. [114]. There was also evidence indicating test-retest reliability and criterion and construct validity [114]. The ECR-I was also found to demonstrate the two-factor structure along with adequate test-retest reliability [115]. Finally, the ECR-J also demonstrated the two-factor structure and results indicated adequate construct validity [116].

The structure of the questionnaire consists of two dimensions, scales, or factors: "attachment-related anxiety" (18 items) and "attachment-related avoidance" (18 items; Cronbach's α Anxiety: 0.94; Cronbach's α Avoidance 0.91) [26]. The combination of the two dimensions gives, as a result, one of the four relational styles according to attachment theory: "Secure," "Dismissing-avoidant," "Preoccupied" and "Fearful-Avoidant" [1, 5, 25–27, 118].

The "Secure" dimension refers to low anxiety and low avoidance in close relationships, which arises due to a positive perception of self and others, where both the self and others show the ability and will to give and receive affection, care, and attention. The "Dismissing-avoidant" dimension refers to low anxiety and high avoidance in close relationships, which arises due to a positive perception of self but negative perception of others, with a vision of oneself as deserving of affection, care, and attention, and of others as incapable or unwilling to give it. The "Preoccupied" dimension refers to high anxiety and low avoidance in close relationships, with the perception of others as capable of showing affection, care, and attention, and of oneself as undeserving of it. The "Fearful-Avoidant" dimension refers to high anxiety and high avoidance in close relationships, with a perception of others as incapable or unwilling of giving affection, care, and attention, and of oneself as undeserving of it [1].

**Parental Bonding Instrument.** The Parental Bonding Instrument–PBI [58] is a self-report questionnaire that provides a retrospective measure of the subject's perception about

mother's and father's behaviors before sixteen. We used the Spanish version of the Parental Bonding Instrument–PBI-S [119, 120] for the actual study. The other versions of PBI used for Italian and Japanese participants were the PBI-I [121] and the PBI-J [122–124], respectively. The PBI-S Spanish (Cronbach's α Care [Affect: 0.93 & Restraint 0.85]; Cronbach's α Overprotection 0.77) [120], the PBI-I Italian (Cronbach's α Care: 0.88 mother & 0.91 father; Cronbach's α Overprotection 0.86 mother & 0.83 father) [121], and PBI-J Japanese show good internal consistency (Cronbach's α Care: 0.94; Cronbach's α Overprotection 0.86) [124]. The PBI-S has been found to demonstrate similar psychometric properties as the PBI and was found to have a two-factor structure, although a three-factor structure appeared to show greater predictive power in relation to affective disorders [120]. The PBI-I also demonstrated high internal consistency and a two-factor solution was also found [121]. The PBI-J used in [122] was demonstrated to show the same two-factor loading in [58] although there have been inconsistent findings of the factorial structure in different Japanese samples [123].

It consists of 50 items on a Likert scale of 3 points (0 = Nothing; 3 = A lot) divided into two sections of 25 items, one for each parent. The PBI classifies the perception of both parental behaviors in two critical dimensions: "Care" and "Overprotection." The "Care" dimension (13 items) refers to affection, emotional warmth, empathy, and closeness. The "Overprotection" dimension (12 items) refers to a parental attitude that stimulates dependence, intrusion, and control of children's behavior even in unnecessary situations, without letting the children act on their own [125].

The combination of the two dimensions gives, as a result, one of the four parental bonding styles: "Optimal bond," "Absent or weak link," "Loving constriction," and "Control without affection." According to Takeda and colleagues [124], mothers' parental bonding experience, as measured by PBI, correlates with an individual's internal working models, affecting the development of their caregiving system toward their offspring. For example, PBI "Optimal bond" (high parental care and low overprotection) is correlated with secure IWM. "Absent or weak link" (low parental care and low overprotection) corresponds to dismissing-avoidant IWM. "Loving constriction" (high care and high overprotection) corresponds to preoccupied IWM. "Control without affection" (low care and high overprotection) corresponds to fearful-avoidant IWM.

## Procedure

Spanish, Italian, and Japanese participants were assessed individually at the European University of Madrid (Spain), University of Trento (Italy), and University of Nagasaki (Japan), respectively. Data were collected from one individual face-to-face session of approximately 40 minutes. In each session, participants completed the questionnaires in the following order: (i) Sociodemographic and clinical survey; (ii) Experience in Close Relationships Scale—ECR; (iii) Parental Bonding Instrument—PBI. Two researchers then coded participants' answers to each questionnaire according to the authors' instructions and developed an ordinal classification to compare the results obtained through the ECR and PBI in the three cultures (Table 2).

The classification was used to define the four relational styles of the ECR, and the eight different relational styles offered by the PBI for each independent sample (Spanish, Italian, and Japanese), using the original factor loadings. Each classification was calculated for the ECR and PBI questionnaire based on the Z scores of each sample. We used the scores of these classifications to compare the actual attachment (ECR) and the perception of the past parental bonding (PBI) in the three cultures.

For the ECR questionnaire classification, the Z scores of each independent group (Spanish N = 100, Italian N = 85, and Japanese N = 120) were calculated for the two ECR factors (F1. Anxiety, F2. Avoidance). Similarly, for the PBI questionnaire classification, we calculated the Z

**Table 2. ECR & PBI ordinal classifications (*ECR Factors: F1-Anxiety, F2-Avoidance, Z scores; **PBI Factors: F1-Care, F2-Overprotection, Z scores).**

| Bowlby, Ainsworth, Main & Solomon Attachment Styles | ECR-Questionnaire | | | PBI-Questionnaire | | |
|---|---|---|---|---|---|---|
| | Level | Attachment style | ECR—Classification * | Level | Bonding style | PBI—Classification ** |
| SECURE | 4 | Secure | F1<+1 SD, F2<+1 SD | 8 | Secure | -1 SD <F1<+1 SD; -1 SD <F2<+1 SD |
| | | | | 7 | Dismissing Low–Secure Low (Low Overprotection) | -1 SD <F1<+1 SD; F2<-1 SD |
| INSECURE AVOIDANT | 3 | Dismissing-avoidant | F2>+1 SD and F1<+1 SD | 6 | Dismissing High—Absent or weak link (Low care & Low Overprotection) | F2<-1 SD; F2<-1 SD |
| INSECURE RESISTANT / ANXIOUS AMBIVALENT | 2 | Preoccupied | F1>+1 SD and F2<+1 SD | 5 | Preoccupied Low–A bit of loving constriction (High Care & Normal Overprotection) | F1>+1 SD; -1 SD <F2< +1 SD |
| | | | | 4 | Preoccupied High—Loving constriction (High care & High Overprotection) | F1>+1 SD; F2>+1 SD |
| | | | | 3 | Preoccupied Very High–A lot of Loving constriction (High Overprotection) | -1 SD <F1<+1 SD; F2> +1 SD |
| INSECURE DISORGANIZED / DISORIENTED | 1 | Fearful avoidant | F1>+1 SD and F2>+1 SD | 2 | Fearful Low—Control without affection (Low Care) | F1<-1 SD; -1 SD <F2< +1 SD |
| | | | | 1 | Fearful High—Control without affection (Low Care & High Overprotection) | F1<-1 SD; F2>+1 SD |

scores of each independent group for the two PBI factors (F1. Mother/Father Care, F2. Mother/Father Overprotection). These classifications allowed the definition of two ECR and PBI main profiles: the balanced (ECR: F1.<+1SD, F2.< +1SD; PBI: -1SD<F1<+1SD, -1SD<F2<+1SD) and the unbalanced ones (ECR: F1.>+1SD, F2.>+1SD; PBI: F1<-1SD, F1>+1SD, F2<-1SD, F2>+1SD), obtained by computing a Z score based on the direct factor score and the standard deviation of the reference group (Spanish, Italian, Japanese; Table 2).

Regarding the ECR classification, for the balanced profile (secure), two conditions must be present: (a) the Z score for the anxiety factor (F1) had to be lower than +1 SD; (b) the Z score for the avoidance factor (F2) had to be lower than +1 SD. If the two conditions were met, the profile was balanced and secure. If not true, the profile was classified as unbalanced under one of the additional three classifications attachment styles (Table 2).

Regarding the PBI classification, for the balanced profile (optimal bond / secure), two conditions must be present: (a) the Z score for the care factor (F1) had to be lower than +1 SD and higher than -1 SD; (b) the Z score for the overprotection factor (F2) had to be lower than +1 SD and higher than -1 SD. If two conditions were met, the profile was classified as balanced and secure. If not true, the profile was classified as unbalanced through one of the additional seven classifications of bonding styles (Table 2).

## Data analysis plan

First, we present the descriptive statistics, Cronbach's alpha, Kolmogorov-Smirnov normality test, and Kruskal Wallis test with Bonferroni Post-Hoc tests for ECR and PBI factors and classifications in the Spanish, Italian, and Japanese groups. To ensure ECR and PBI factors measures are comparable, we run a multigroup factorial analysis (CFA) to test the measurement invariance and reduce possible biases in cross-cultural research [126]. Four models are explored, establishing a progressive series of restrictions on model parameters to assess the invariance between groups [127, 128]. To assess the differences between the models, we use the variations of the CFI indices (ΔCFI) and RMSEA (ΔRMSEA). The strong invariance is assumed when ΔCFI ≤ 0.01 and ΔRMSEA are ≤ 0.015 [129], while partial invariance is assumed when ΔCFI ≤ 0.01 or ΔRMSEA is ≤ 0.015 [130].

ECR and PBI gender differences are explored through the Mann-Whitney U test for independent groups. To investigate correlations between age, ECR, PBI factors, and classifications in the three groups, we calculated Spearman's r correlations. For hypotheses 1 and 2, we performed the Kruskal-Wallis test with Bonferroni Post-Hoc tests to explore differences between ECR secure and insecure classifications, anxiety, and avoidance factors scores. For hypothesis 3, we investigated correlations and differences between ECR and PBI factors through Spearman's r correlation and Kruskal-Wallis with Bonferroni Post-Hoc test.

## Results

### Descriptive statistics

Descriptive statistics, internal consistency coefficients, and Kolmogorov-Smirnov normality test for the ECR and PBI factors in the Spanish, Italian and Japanese groups are shown in Table 3. Non-parametric statistics were reported for study variables which were not normally distributed.

**Table 3. Descriptive statistics for the ECR and PBI scales.**

| | Group | Average (SD) | Cronbach's alpha (α) | Kolmogorov-Smirnov Significance | Kruskal Wallis Test | Bonferroni post-hoc | |
|---|---|---|---|---|---|---|---|
| ECR–Anxiety | Spain | 61.18 (15.61) | 0.851 | **0.20** | 7.57 (df 2), $p = 0.02^*$ | Spain & Italy | $p = 1.00$ |
| | Italy | 63.26 (20.58) | 0.908 | **0.20** | | Spain & Japan | $p = 0.05^*$ |
| | Japan | 55.40 (21.10) | 0.927 | $< 0.01$ | | Italy & Japan | $P = 0.03^*$ |
| ECR–Avoidance | Spain | 54.19 (14.61) | 0.890 | **0.15** | 45.13 (df 2), $p < 0.01^{**}$ | Spain & Italy | $p = 0.03^*$ |
| | Italy | 47.78 (15.66) | 0.883 | 0.03 | | Spain & Japan | $p < 0.01^{**}$ |
| | Japan | 64.48 (18.71) | 0.894 | 0.04 | | Italy & Japan | $p < 0.01^{**}$ |
| ECR-Classification | Spain | 3.52 (0.76) | | $< 0.01$ | 0.53 (df 2), $p = 0.77$ | Spain & Italy | $p = 1.00$ |
| | Italy | 3.49 (0.87) | | $< 0.01$ | | Spain & Japan | $p = 1.00$ |
| | Japan | 3.53 (0.91) | | $< 0.01$ | | Italy & Japan | $p = 1.00$ |
| PBI-Mother Care | Spain | 27.41 (6.66) | 0.882 | $< 0.01$ | 125.46 (df 2), $p < 0.01^{**}$ | Spain & Italy | $p < 0.01^{**}$ |
| | Italy | 17.01 (2.45) | 0.943 | $< 0.01$ | | Spain & Japan | $p < 0.01^{**}$ |
| | Japan | 17.40 (3.31) | 0.810 | $< 0.01$ | | Italy & Japan | $p = 1.00$ |
| PBI-Mother Overprotection | Spain | 12.16 (8.09) | 0.928 | $< 0.01$ | 71.92 (df 2), $p < 0.01^{**}$ | Spain & Italy | $p < 0.01^{**}$ |
| | Italy | 15.85 (4.02) | 0.720 | 0.05 | | Spain & Japan | $p < 0.01^{**}$ |
| | Japan | 19.06 (3.67) | 0.854 | $< 0.01$ | | Italy & Japan | $p < 0.01^{**}$ |
| PBI-Mother- Classification | Spain | 6.80 (1.63) | | $< 0.01$ | 22.45 (df 2), $p < 0.01^{**}$ | Spain & Italy | $p = 1.00$ |
| | Italy | 6.89 (1.64) | | $< 0.01$ | | Spain & Japan | $p < 0.01^{**}$ |
| | Japan | 7.48 (1.33) | | $< 0.01$ | | Italy & Japan | $p < 0.01^{**}$ |
| PBI-Father-Care | Spain | 22.62 (8.32) | 0.913 | **0.09** | 7.25 (df 2), $p = 0.03^*$ | Spain & Italy | $p = 0.02^*$ |
| | Italy | 20.28 (5.45) | 0.938 | 0.02 | | Spain & Japan | $p = 0.68$ |
| | Japan | 21.77 (5.04) | 0.896 | $< 0.01$ | | Italy & Japan | $p = 0.68$ |
| PBI-Father-Overprotection | Spain | 9.16 (7.18) | 0.918 | $< 0.01$ | 19.44 (df 2), $p < 0.01^{**}$ | Spain & Italy | $p < 0.01^{**}$ |
| | Italy | 11.66 (4.29) | 0.938 | $< 0.01$ | | Spain & Japan | $p < 0.01^{**}$ |
| | Japan | 11.48 (4.03) | 0.870 | $< 0.01$ | | Italy & Japan | $p = 1.00$ |
| PBI-Father- Classification | Spain | 6.32 (2.22) | | $< 0.01$ | 18.20 (df 2), $p < 0.01^{**}$ | Spain & Italy | $p = 1.00$ |
| | Italy | 6.58 (1.93) | | $< 0.01$ | | Spain & Japan | $p < 0.01^{**}$ |
| | Japan | 7.18 (1.62) | | $< 0.01$ | | Italy & Japan | $p < 0.01^{**}$ |

$^*$ $p < 0.05$;

$^{**}$ $p < 0.01$.

To explore the measurement invariance of ECR and PBI factors in the three groups, a multigroup factorial analysis (CFA) was run. The ΔCFI scores (invariance of CFA models) suggest that the ECR (F1. Anxiety; F2. Avoidance) and PBI (F1. Care; F2. Overprotection) bifactorial structures do not fit well, underlying a factor model variance in the three groups (Table 4). On the other hand, the ΔRMSEA (equivalence of metric variance) shows metric invariance in the three groups, suggesting that each item contributed to the latent construct to a similar degree across the Spanish, Italian and Japanese groups (Table 4). In conclusion, the results indicate partial invariance of the ECR metrics in the three groups.

Regarding the ECR questionnaire factors (F1. Avoidance, F2. Anxiety) and the ECR classification (1. Fearful-Avoidant, 2. Preoccupied, 3. Dismissing-avoidant, 4. Secure), results show significant gender differences only in the Italian sample, with higher avoidance scores observed in men than in women (F1. Avoidance; U = 989.50; $p$ = 0.03; Males M = 52.23; Females M = 45.81).

Regarding the PBI questionnaire factors (F1-Mother/Father Care, F2-Mother/Father Overprotection), results show no significant gender differences for each group and for the whole

**Table 4. Multigroup confirmatory factorial analysis (CFA).**

| | ① Model | X² (df) | X²/DF | ② CFI | ③ RMSEA (IC;90%) | Comparison | ④ ΔX² | ⑤ ΔCFI * | ⑥ ΔRMSEA* |
|---|---|---|---|---|---|---|---|---|---|
| **ECR (F1. Anxiety; F2. Avoidance)** | M1—Unconstrained (baseline) | 4038,56 (1742) | 2,27 | 0,64 | 0,065 (0,062; 0,067) | | | | |
| | **M2—Measurement weights** | 4407,88 (1850) | 2,83 | 0,59 | 0,068 (0,065;0,070) | M2 vs M1 | 369,32 (108), p < 0,01 | -0,05 | 0,003 |
| | M3—Measurement intercepts | 5582,30 (1922) | 2,90 | 0,42 | 0,079 (0,077;0,082) | M3 vs M2 | 1174,42 (72), p < 0,01 | -0,17 | 0,011 |
| | M4—Measurement residuals | 6079,85 (1988) | 3,04 | 0,36 | 0,082 (0,080; 0,085) | M4 vs M3 | 497,55 (66), p < 0,01 | -0,06 | 0,003 |
| **PBI-M (F1. Care; F2. Overprot.)** | M1—Unconstrained (baseline) | 1844,46 (825) | 2,24 | 0,73 | 0,064 (0,060; 0,068) | | | | |
| | M2—Measurement weights | 1996,61 (871) | 2,92 | 0,71 | 0,065 (0,062; 0,069) | M2 vs M1 | 152,15 (46), p < 0,01 | -0,02 | 0,001 |
| | M3—Measurement intercepts | 2397,89 (921) | 2,60 | 0,62 | 0,073 (0,069; 0,076) | M3 vs M2 | 431,28 (50), p < 0,01 | -0,09 | 0,008 |
| | M4—Measurement residuals | 2720,42 (975) | 2,79 | 0,55 | 0,077 (0,074; 0,080) | M4 vs M3 | 322,53 (54), p < 0,01 | -0,07 | 0,004 |
| **PBI-F (F1. Care; F2. Overprot.)** | M1—Unconstrained (baseline) | 1767,81 (825) | 2,14 | 0,77 | 0,062 (0,058; 0,065) | | | | |
| | M2—Measurement weights | 1912,63 (871) | 2,20 | 0,75 | 0,063 (0,059; 0,067) | M2 vs M1 | 144,83 (46), p < 0,01 | -0,02 | 0,001 |
| | M3—Measurement intercepts | 2381,30 (921) | 2,59 | 0,65 | 0,072 (0,069; 0,076) | M3 vs M2 | 468,66 (50), p < 0,01 | -0,1 | 0,009 |
| | M4—Measurement residuals | 2642,263 (975) | 2,71 | 0,60 | 0,075 (0,072; 0,079) | M4 vs M3 | 260,96 (54), p < 0,01 | -0,05 | 0,003 |

Values indicating metric invariance are highlighted in grey.

① M1: Unconstrained invariance model (baseline) with free estimation of factor loadings, intercepts, and error variance in each group. M2: Measurement weights model (metric invariance) with equal factor loadings in the groups. M3: Measurement intercepts model (scalar invariance) with equal intercepts and factor loadings in the groups. M4: Measurement residuals model (strict invariance) with equal intercepts, factor loadings, and error variance in the groups.

② CFI—Comparative Fit Index; CFI ≥ 0.95 the model fits the sample [131].

③ RMSEA—Root mean squared error of approximation; RMSEA ≤ 0.05 the model fits the sample [131].

④ ΔX² Invariance assumed when p> 0.05.

⑤ ΔCFI (between-group invariance of CFA models). Factor model invariance assumed when p< 0.01.

⑥ ΔRMSEA (equivalence of metric variance). Metric invariance assumed when p< 0.015.

* Strong invariance is assumed when ΔCFI ≤ 0.01 and ΔRMSEA ≤ 0.015 [129]. Partial invariance is assumed when ΔCFI ≤ 0.01 or ΔRMSEA ≤ 0.015 [130].

sample. On the other hand, results show significant gender differences in the PBI classification only across the whole sample (Italy, Spain & Japan), where men had higher scores in both the PBI-Mother classification (U = 13109; $p$ = 0.02; Males M = 7.29; Females M = 6.87) and the PBI-Father classification (U = 13011; $p$ = 0.04; Males M = 7.0; Females M = 6.43). This trend was not observed in any of the individual groups (PBI-Mother classification: Spain U = 1076; $p$ = 0.19; Italy U = 663; $p$ = 0.27; Japan U = 1439; $p$ = 0.53; PBI-Father classification: Spain U = 1055,5; $p$ = 0.15; Italy U = 616.5; $p$ = 0.12; Japan U = 1347; $p$ = 0.21).

Only the correlation between age and PBI Mother classification was significant in the Japanese sample (Spearman's r = 0.201; $p$ = 0.03) (Table 5).

As shown in Table 6, the dimensions of ECR anxiety and avoidance are negatively related to the ECR classification in the three groups. The greater the attachment-related anxiety and avoidance are, the lower the security reflected by the classification (Table 2). On the other hand, the results show that PBI-Mother/Father Care and Overprotection dimensions merge in a sophisticated eight-point index (Table 2) in the PBI Mother and Father classifications (Table 6), making it lose direct correlation with the two PBI factors in the three groups.

## Hypothesis 1

In the current sample, the more frequent attachment style in the three groups (Spain, Italy, and Japan; Fig 1) is the secure one (68–74%), followed by dismissing-avoidant (11%-16%), preoccupied (8%-16%) and finally fearful-avoidant (0%-7%). There are no significant differences between the three groups in terms of ECR classification scores calculated on each group's mean and standard deviation (Kruskal Wallis Test = 0.53, df = 2, $p$ = 0.77; Table 3). Hence, the results support Hypothesis 1.

On the other hand, in the three groups, the most frequent maternal and paternal bonding style was the secure one (Figs 2 and 3). The results show Japanese participants recall a significantly more secure bonding with both mothers and fathers than Italians and Spaniards (PBI-Mother classification: $\chi^2$ = 31.88; df 10; $p$ < 0.01; PBI-Father classification: $\chi^2$ = 42.43; df 10; $p$ < 0.01).

## Hypothesis 2

The results from the Kruskal Wallis Test with Bonferroni Post-Hoc correction support Hypothesis 2. For the Mediterranean cultures, the most prevalent insecure style is the preoccupied one (high attachment-related anxiety and low attachment-related avoidance), while for Japanese participants, the most prevalent insecure style is the dismissing-avoidant one (high

**Table 5. Spearman's r correlations between age, ECR and PBI factors.**

|  | Spain (N 100) | Italy (N 85) | Japan (N 120) |
|---|---|---|---|
| ECR-Anxiety & Age | r = -0.137 ($p$ = 0.173) | r = -0.078 ($p$ = 0.479) | r = -0.159 ($p$ = 0.082) |
| ECR-Avoidance & Age | r = 0.005 ($p$ = 0.963) | r = 0.021 ($p$ = 0.847) | r = -0.120 ($p$ = 0.192) |
| ECR-Classification & Age | r = 0.166 ($p$ = 0.099) | r = 0.159 ($p$ = 0.145) | r = 0.037 ($p$ = 0.689) |
| PBI-Mother Care & Age | r = -0.109 ($p$ = 0.281) | r = 0.021 ($p$ = 0.846) | r = -0.009 ($p$ = 0.925) |
| PBI-Mother Overprotection & Age | r = -0.037 ($p$ = 0.711) | r = 0.167 ($p$ = 0.127) | r = -0.141 ($p$ = 0.125) |
| PBI-Mother-Classification & Age | r = -0.003 ($p$ = 0.978) | r = -0.043 ($p$ = 0.696) | r = 0.201 ($p$ = 0.028) * |
| PBI-Father-Care & Age | r = 0.091 ($p$ = 368) | r = -0.041 ($p$ = 0.711) | r = -0.016 ($p$ = 0.866) |
| PBI-Father-Overprotection | r = -0.051 ($p$ = 0.613) | r = 0.010 ($p$ = 0.930) | r = 0.039 ($p$ = 0.673) |
| PBI-Father-Classification & Age | r = -0.030($p$ = 0.765) | r = -0.051 ($p$ = 0.644) | r = -0.016 ($p$ = 0.864) |

* $p$ < 0.05.

**Table 6. Spearman's r correlations between ECR and PBI factors and classifications.**

| | Spain (N = 100) | Italy (N = 85) | Japan (N = 120) |
|---|---|---|---|
| Anxiety ECR & ECR-Classification | r = -0,54 ($p < 0.01$) ** | r = -0,59 ($p < 0.01$) ** | r = -0,41 ($p < 0.01$) ** |
| Avoidance ECR & ECR-Classification | r = -0,32 ($p < 0.01$) ** | r = -0,52 ($p < 0.01$) ** | r = -0,50 ($p < 0.01$) ** |
| PBI-Mother Care & PBI-Mother-Classification | r = 0,03 ($p = 0.76$) | r = -0,13 ($p = 0.22$) | r = -0,001 ($p = 0.99$) |
| PBI-Mother Overprotection & PBI-Mother-Classification | r = -0,20 ($p = 0.05$) * | r = -0,23 ($p = 0.03$) * | r = -0,11 ($p = 0.23$) |
| PBI-Father Care & PBI-Father-Classification | r = -0,10 ($p = 0.92$) | r = 0,001 ($p = 0.99$) | r = 0,13 ($p = 0.15$) |
| PBI-Father Overprotection & PBI-Father-Classification | r = -0,11 ($p = 0.27$) | r = -0,18 ($p = 0.11$) | r = -0,10 ($p = 0.27$) |

* $p < 0.05$;

** $p < 0.01$.

attachment-related avoidance and low attachment-related anxiety) (Fig 1 and Table 3). Italians scored higher than Spaniards and Japanese for ECR-Anxiety, with a significant difference found only between Italy and Japan (Kruskal Wallis Test 7.57; df 2; $p = 0.02$; Table 3). On the other hand, the Japanese scored the highest values for the ECR-Avoidance, followed by Spaniards and Italians, with a significant difference between the three groups ($\chi 2 = 43.13$; $p < 0.01$; Table 3).

### Hypothesis 3

The results partially support Hypothesis 3, which states that ECR attachment-related avoidance in Japanese culture is correlated with higher recalled parental overprotection. In contrast, in the Spanish culture, higher attachment-related anxiety is correlated with higher recalled parental care. In the Italian sample, there is no significant relationship between current adult close relationships and recalled parental bonding (Figs 4–6).

Attachment-related avoidance in the Japanese group was correlated with higher levels of recalled maternal and paternal overprotection and lower levels of recalled paternal care (Fig 4). Moreover, attachment-related anxiety is related to more substantial experience of recalled paternal overprotection and a lower recalled paternal care (Fig 4). In the Spanish group, higher attachment-related avoidance is related to a lower recalled maternal and paternal care (Fig 5).

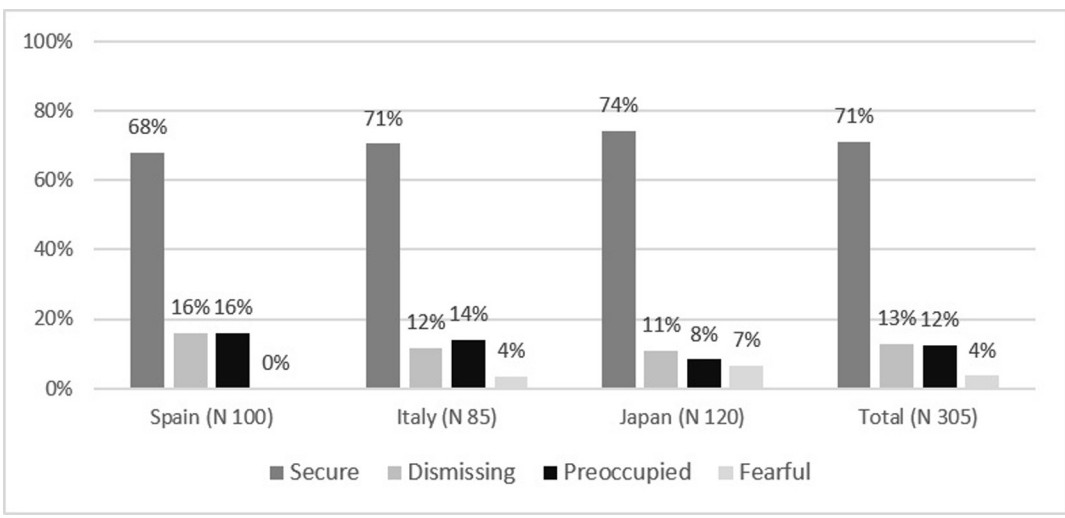

**Fig 1. Frequencies of attachment styles derived using the ECR classification by groups.**

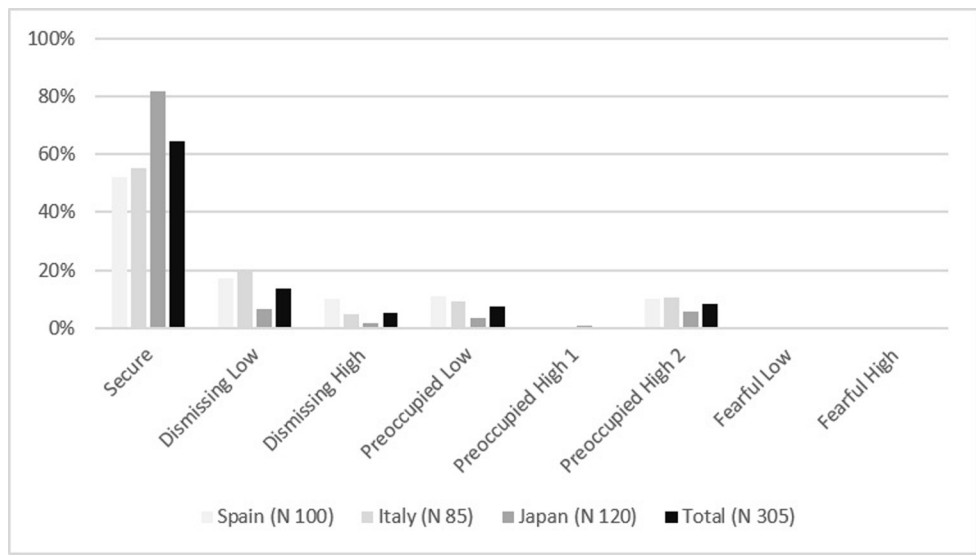

**Fig 2. Maternal bonding frequencies among groups (PBI-Mother classification).**

Attachment-related anxiety was not correlated with recalled parental bonding in the Spanish group (Fig 5). Finally, in the Italian sample, there were no significant correlations between PBI measures and the ECR dimensions of attachment-related anxiety and avoidance (Fig 6).

For PBI Mother factors, the Spaniards scored the highest values for the care factor, followed by the Italians and Japanese, with a significant difference between Spain versus Italy and Japan (Kruskal Wallis Test = 125.46; df 2; $p < 0.01$; Table 3). On the other hand, the Japanese scored the highest values for the overprotection factor, followed by Italians and Spaniards, with a significant difference between the three groups (Kruskal Wallis Test = 71.92; df 2; $p < 0.01$; Table 3).

Regarding the PBI Father factors, the Spaniards scored the highest values for care, followed by Japanese and Italians, with a significant difference only between Spain and Italy (Kruskal

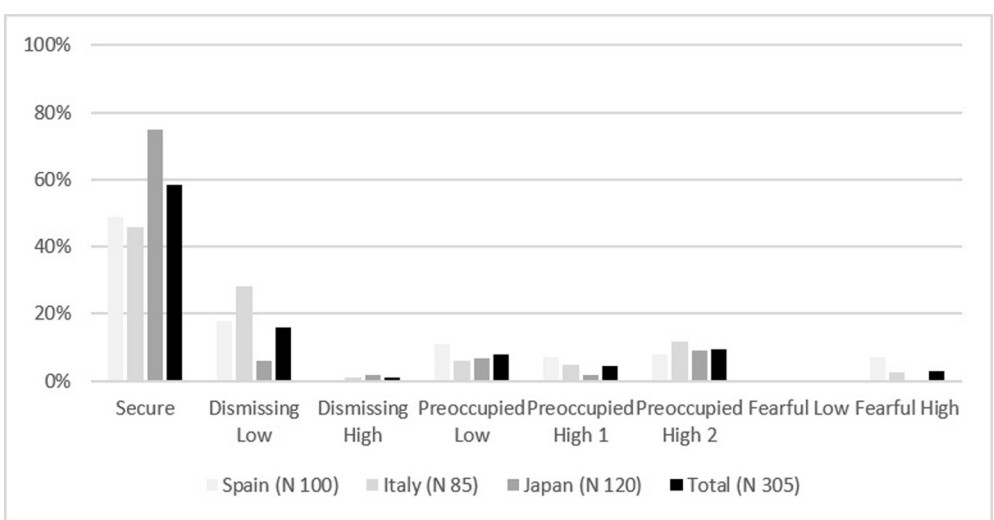

**Fig 3. Paternal bonding frequencies among groups (PBI-Father classification).**

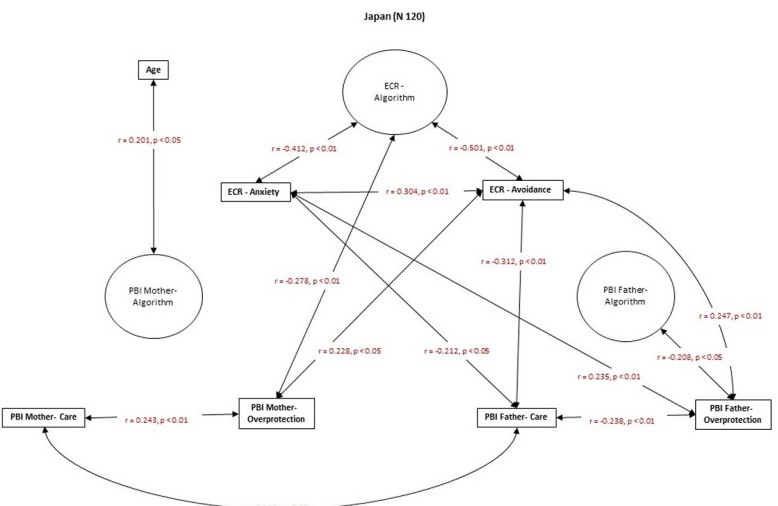

**Fig 4. Spearman's r correlations between ECR and PBI in the Japanese group.**

Wallis Test = 7.25, df 2; $p$ = 0.03; Table 3). Finally, Italians and Japanese scored similar highest values for the overprotection factor, with significant differences with Spaniards (Kruskal Wallis Test = 19.44; df 2; $p$ < 0.01; Table 3).

## Discussion

The present study investigates the relationship between adult attachment style and recalled parental bonding in Spanish, Italian, and Japanese cultures. For this purpose, we administered the validated versions of the Experience in Close Relationship (ECR) and the Parental Bonding (PBI) questionnaires to three hundred and five participants. We calculated two classifications to compare the three groups according to their inner characteristics. For the ECR, the classification (four-point index) showed to be a valuable and valid instrument strongly correlated with the two internal factors. For the PBI, the classification was more sophisticated (eight-

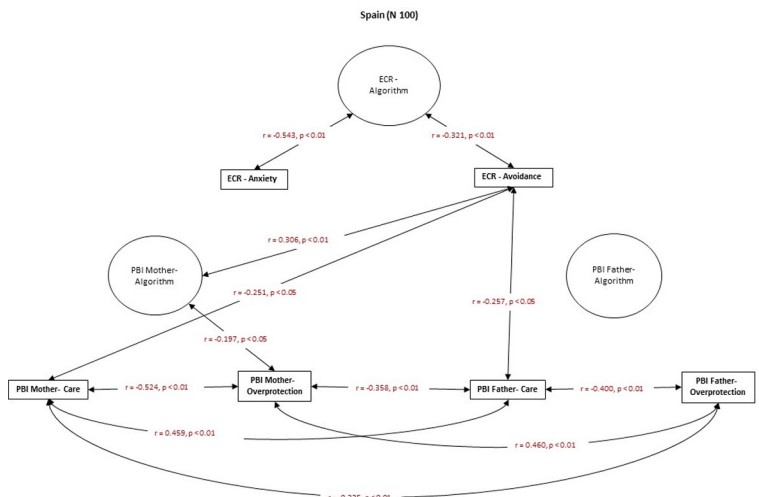

**Fig 5. Spearman's r correlations between ECR and PBI in the Spanish group.**

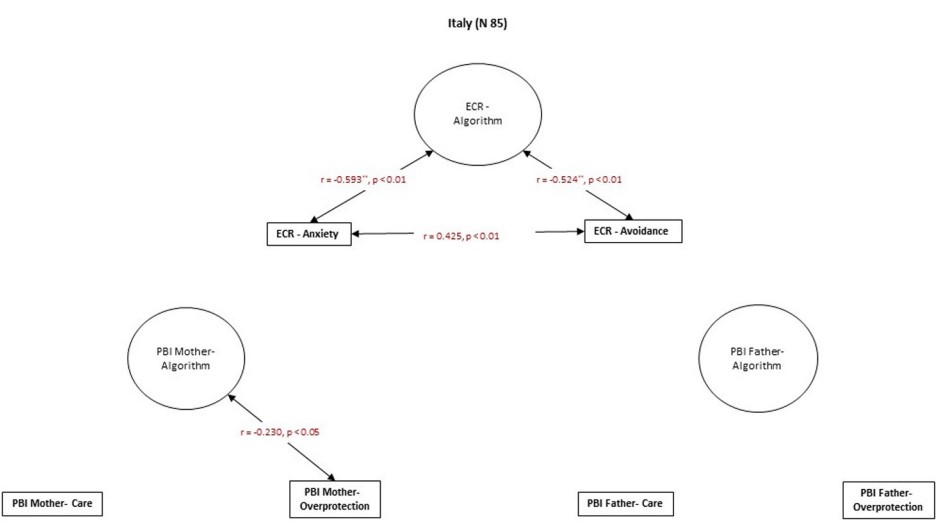

**Fig 6. Spearman's r correlations between ECR and PBI in the Italian group.**

point index), making it lose direct correlations with the two PBI factors, so the data analysis focused on the questionnaire's factors. The main difference between the ECR and PBI classifications is that, in the first case, both the two ECR factors (attachment-related anxiety and attachment-related avoidance) converge negatively at a higher level of attachment security. On the other hand, the two PBI factors (care and overprotection) measure opposite variables in the optimal bond. So, a high parental bonding classification index depends on high care and low overprotection scores.

Results show significant gender differences only in the ECR factors for the Italian group (with higher attachment-related avoidance scores in men) and a positive correlation with age was only observed with the PBI Mother classification in the Japanese group. Despite no significant differences in the secure and insecure attachment due to gender [15, 27], results indicate a specific attachment tendency depending on the culture, which seems to be the case in Italy, where men tend to be more avoidant, and women are more preoccupied [60–62]. On the other hand, the correlation between age and the PBI mother classification in the Japanese group indicates a more optimal maternal bond recalled by participants as age advances, considering a range between 19 and 37 years.

## Hypothesis 1

The results show no significant differences between the Spaniards, Italians, and Japanese regarding the frequency of secure and insecure attachment style, according to their cultural characteristics assumed through the ECR classification. In all cases, the most frequent adult attachment style is the secure one, ranging between 68% and 74% of the participants in the three groups, followed by the dismissing-avoidant (11%-16%), the preoccupied style (8%-16%), and finally, the fearful-avoidant style (0%-7%). These results are in line with distributions obtained in other studies [70, 74, 87–90, 93, 94]. On the other hand, although the most frequent maternal (52–82%) and paternal (46%-75%) bonding styles fall at a secure level (optimal bond) in the three cultures, in most cases, Japanese recall a significantly more optimal bonding with mother and father than Italians and Spaniards.

The behavioral system of attachment is an evolutionary, biological, and innate mechanism of the species, which is expressed through the interaction with the environment, designed to

increase the individual's probability of survival and reproductive success [28, 29]. The secure attachment style is functional to the individual's adaptation and the exploration of the social environment, according to the canons of each specific culture. For this reason, it is understandable that, in the non-clinical population, secure attachment is the most frequent in different cultures, as a universal component of human genetic programming that is expressed through the interaction with the close relationships and relational environment [29, 132]. On the other hand, insecure styles tend to be more rigid, less flexible to the variability of human relational experiences, and therefore more limited in fitting the new experience with the previous expectations. For this reason, it is understandable that the most prevalent attachment styles in the clinical population are the insecure ones [38, 39, 42–44, 133, 134].

## Hypothesis 2

Even though research supports that attachment styles are universal and analogous across cultures [135], the present study supports this notion for insecure styles. Previous results are already inconsistent [62], and probably the universality of attachment styles cannot be assigned to insecure attachment ones across cultures.

Independently from the culture, we expected to find that the preoccupied attachment style (high attachment-related anxiety and low attachment-related avoidance) was the more prevalent of the insecure attachments [48, 83, 89, 90, 94, 96, 97]. However, results show that Japanese participants tended more towards attachment-related avoidance and less towards attachment-related anxiety (dismissing-avoidant style). In contrast, Mediterranean cultures (Italy and Spain) tended more towards attachment-related anxiety and less towards attachment-related avoidance (preoccupied style).

Previous research states that in Japan, the prevailing cultural model is collectivism, based on a lifetime relationship of subordination from offspring to parents to ensure familial stability [48, 75–77]. The relationship's standards in Japan delve on lasting loyalty, conflict avoidance, pragmatism, harmony, and the actual study reflects it [72, 136]. Avoidant people tend to display behaviors such as withdrawal and inhibition to avoid conflict in the relationship and maintain harmony [49]. So, the higher avoidance of the Japanese group probably shows a cultural model of sacrificing the individual demands and subordination of personal requirements in favor of the benefit of the relationship.

On the other hand, Western Mediterranean cultures like Italian and Spanish are more focused on socialization and extra-familiar interdependence [80, 82, 112], tending less towards avoidance and more towards anxiety. Italians tend to focus on social interactive and affective exchanges related to enjoying life [83–86], while Spaniards tend to focus on interactive social exchanges related to being a good citizen and family member [81, 83, 112]. In any case, in adulthood, Mediterranean countries are based on romantic interactions, verbal connection, mutual attraction, intimacy, and sexual intercourse [72, 136]. This yearning for intimacy and perseveration in staying in a relationship is related to higher levels of anxiety in terms of expectations of oneself and others [113]. It is probably why Mediterranean countries showed higher levels of attachment-related anxiety in our sample than Japanese ones. East Asian and Mediterranean cultures consider romanticism necessary before marriage, but Western countries maintain this value, even after child-rearing [103–105].

## Hypothesis 3

The results partially support the third hypothesis, showing a correlation between the attachment-related avoidance dimension and the memory of parental overprotection in the Japanese group.

In the three countries, the most frequent memory of parental bonding style was an optimal bond where the Japanese recalled a more optimal bond with both parents than the Italians and Spaniards. Significant gender differences are observed in the total sample in terms of the classifications of attachment styles. In general, men have a more optimal memory of affective bonding with both the father and the mother than women.

Japanese were more anchored to maternal and paternal overprotection, related to more avoidance in existing close relationships. Spaniards mildly distanced present relationships from recalled parental bonding, connecting less current avoidance to primary parental care. Italians separated present close relationships from the memory of parental bonding.

A higher mother-child dependence is expected in Japan [72, 78, 79]. It probably feeds a trend to avoid closeness [27, 59, 108, 109, 111]. This trend is unshared with Western cultures, which are more care-oriented [106, 107], underling the controversy about whether attachment principles are universal or culturally dependent [73]. Western cultures, such as Italian and Spanish ones, consider "parental sensitivity" as the ability to perceive and respond effectively to the child's cues and requests for help [91]. For Eastern cultures, like the Japanese one, "parental sensitivity" also consists of anticipating the child's needs and preventing the child's calls for help, which requires maximum mother-child dependency [69, 72].

Moreover, the current study did not observe significant correlations between attachment-related anxiety and parental care in the Spanish or the Italian group. It is coherent with previous research on Italian [137], European, North American, and Israeli populations [138, 139]. In the Spanish group, somewhat lower attachment-related avoidance is related to a higher recalled maternal and paternal care. Different studies demonstrated that attachment-related anxiety is negatively correlated to secure attachment style and can partially mediate the relationship between memories of low care and self-reported obsessive beliefs [140, 141]. On the other hand, higher attachment-related avoidance has been found in people with depression and unmarried people with obsessive-compulsive disorder [142]. So, the relationship between lower avoidant tendency and higher recalled parental care observed in this study for Spaniards can be considered a protective factor toward social exploration.

Maternal and paternal overprotection and the current close relationship style appeared firmer in Japanese culture, with a more enduring connection between alive attachment style and recalled parental bonding representations. In Spanish culture, the memory of past parental care is associated with less avoidant models of close relationships. Finally, the past parental bonding models and the present relationships appear dissociated in Italian culture.

## Limitations

The primary study limitation is the definition of the PBI classification. In trying to achieve tighter categories, the PBI classification became too complex with an 8-point scale interconnecting parental care and overprotection dimensions in a more complex way than desired. It produced the loss of direct correlations with the two PBI factors. Probably, a simpler PBI classification with a 4-point scale such as the ECR classification would have given us more information to compare the three samples more closely.

Another study limitation was the gender differences found in the Italian sample for the ECR factors and classification. Results show Italian men reached higher scores in ECR avoidance than women, which could have been due to the gender imbalance and small size of the Italian group (N85, 31% men, 69% women) compared to the Spanish samples (N100, 50% men, 50% women) and Japanese (N120, 70% men, 30% women).

The study screened non-clinical participants through a brief sociodemographic questionnaire which only asked participants whether they had been diagnosed with psychopathology

or not and whether they were on any psychiatric medication. The study would have benefited from a deeper psychological assessment, allowing greater accuracy in finding false negatives.

Finally, the results in the present study should be interpreted with caution due to the relatively limited sample collected from each country. The participants were also sampled from university students in the three countries, which limits their generalisability to their respective cultures. Although the individualism-collectivism framework and the assumptions of Western cultures as individualistic and Eastern cultures as collectivistic is often applied in cross-cultural research, it is important to note that individualism-collectivism is continuous rather than a dichotomous categorisation and such cultural generalizations may not necessarily reflect each country as a whole. As demonstrated in a number of studies, countries themselves may not be culturally homogenous [143, 144]. For example, there are significant regional differences between the Northern and Southern regions of Italy [145] and the sample in the current study was collected from a Northern autonomous province of Italy. Hence, future studies can also look to examine adult attachment within intranational subcultures as well.

## Conclusion

Regarding adult attachment styles, in Italy, Spain, and Japan, the most frequent adult attachment style observed in the present study was the secure one, followed by the dismissing-avoidant, the preoccupied, and the fearful-avoidant style. In this regard, men in Italy were observed to be more avoidant than women.

The most frequent insecure attachment style in Japan was dismissing-avoidant (high attachment-related avoidance and low attachment-related anxiety), while in the Mediterranean countries (Italy and Spain), it was the preoccupied attachment style (low attachment-related avoidance and high attachment-related anxiety). In Japan, it is due, probably, to a familiar model defined by interdependence and subordination of personal requirements in favor of the benefit of the relationship. In Spain and Italy, it is due, probably, to a social model focused on socialization and extra-familiar interdependence, affective exchange, and interpersonal intimacy.

These results highlight the need to consider cultural factors when exploring representations of close relationships in both non-clinical and clinical populations. Different cultural models influence adult attachment representations, placing weight on attachment-related avoidance, attachment-related anxiety, care, and overprotection in infant and adult relationships. On the one hand, like in Japan, the cultural model in close relationships is more oriented to intra-familiar interdependency and overprotection, anticipating the other's needs and requests for help. It can drive conflict avoidance, pragmatism, loyalty, and harmony in adult relationships. On the other hand, like in Italy and Spain, the cultural model in close relationships is more extra-familiar oriented, waiting for the others to express their needs and requests for help. It can drive anxiety, searching for emotional availability, love, and responsiveness due to expectations of receiving and giving care.

## Acknowledgments

All participants are gratefully acknowledged. We would like to acknowledge members of the Social Affective Neuroscience Lab at NTU for their assistance in the completion of this project.

## Author Contributions

**Conceptualization:** Maria Alejandra Koeneke Hoenicka, Oscar López-de-la-Nieta, Kazuyuki Shinohara, Gianluca Esposito, Giuseppe Iandolo.

**Formal analysis:** Maria Alejandra Koeneke Hoenicka, Oscar López-de-la-Nieta, Kazuyuki Shinohara, Gianluca Esposito, Giuseppe Iandolo.

**Investigation:** Maria Alejandra Koeneke Hoenicka, Oscar López-de-la-Nieta, Kazuyuki Shinohara, Gianluca Esposito, Giuseppe Iandolo.

**Methodology:** Maria Alejandra Koeneke Hoenicka, Oscar López-de-la-Nieta, Kazuyuki Shinohara, Gianluca Esposito, Giuseppe Iandolo.

**Writing – original draft:** Maria Alejandra Koeneke Hoenicka, Oscar López-de-la-Nieta, Kazuyuki Shinohara, Michelle Jin Yee Neoh, Gianluca Esposito, Giuseppe Iandolo.

**Writing – review & editing:** Maria Alejandra Koeneke Hoenicka, Oscar López-de-la-Nieta, José Luis Martínez Rubio, Michelle Jin Yee Neoh, Dagmara Dimitriou, Gianluca Esposito, Giuseppe Iandolo.

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
