## [Decision Letter · Decision Letter 0]

27 Sep 2022

PONE-D-22-15135Parental bonding in retrospect and adult attachment style: A comparative study between Spanish, Italian and Japanese culturesPLOS ONE

Dear Dr. Esposito,

Thank you for submitting your manuscript to PLOS ONE. After careful consideration, we feel that it has merit but does not fully meet PLOS ONE’s publication criteria as it currently stands. Therefore, we invite you to submit a revised version of the manuscript that addresses the points raised during the review process.

 Please note that we have only been able to secure a single reviewer to assess your manuscript. We are issuing a decision on your manuscript at this point to prevent further delays in the evaluation of your manuscript. Please be aware that the editor who handles your revised manuscript might find it necessary to invite additional reviewers to assess this work once the revised manuscript is submitted. However, we will aim to proceed on the basis of this single review if possible. Please submit your revised manuscript by Nov 07 2022 11:59PM. If you will need more time than this to complete your revisions, please reply to this message or contact the journal office at plosone@plos.org. Please include the following items when submitting your revised manuscript:A rebuttal letter that responds to each point raised by the academic editor and reviewer(s). You should upload this letter as a separate file labeled 'Response to Reviewers'.A marked-up copy of your manuscript that highlights changes made to the original version. You should upload this as a separate file labeled 'Revised Manuscript with Track Changes'.An unmarked version of your revised paper without tracked changes. You should upload this as a separate file labeled 'Manuscript'.

We look forward to receiving your revised manuscript.

Kind regards,

Callam Davidson

Editorial Office

PLOS ONE

Journal Requirements:

Please ensure that the study is reported according to the STROBE guideline, and include the completed STROBE checklist as Supporting Information. Please add the following statement, or similar, to the Methods: "This study is reported as per the Strengthening the Reporting of Observational Studies in Epidemiology (STROBE) guideline (S1 Checklist)."

Reviewers' comments:

Reviewer's Responses to Questions

**Comments to the Author**

1. Is the manuscript technically sound, and do the data support the conclusions?

Reviewer #1: No

2. Has the statistical analysis been performed appropriately and rigorously? 

Reviewer #1: No

3. Have the authors made all data underlying the findings in their manuscript fully available?

Reviewer #1: Yes

4. Is the manuscript presented in an intelligible fashion and written in standard English?

Reviewer #1: Yes

5. Review Comments to the Author

Reviewer #1: The current manuscript makes a good contribution to the adult attachment framework. The study investigates the relations of parental bonding and adult attachment style in three different samples from 3 countries. The previous findings of the link between perceived parental bonding and attachment style have been conflicting. The current study attempts to clarify the relationships. Although the current study presents meaningful empirical cross-cultural data, I find that the information presented in the methodology, results, and discussion sections should be revised to improve cross-cultural aspects of the study before accepting for publication. Thus, my decision is to revise and resubmit. My specific feedbacks are as follows.

Abstract. I find the final sentence in this section “However, it does not exclude activation of original Internal Working Models to contexts in which the individual interprets a danger to the relationship” does not best represent many good arguments the authors have in the discussion section. The replacement of this sentence would improve the section.

Introduction. This section is well-written overall. The review of the previous findings is extensive. The rationale of the study is sound.

Method and results. These sections need some clarification to strengthen the cross-cultural research approach. The description of measurements used in the study needs additional information about the examination of measurement invariance to ensure the measures used are comparable across cultures. Please see Lacko et al. (2022) https://doi.org/10.1177/10693971211068971 for guideline on statistical analyses to reduce biases in cross-cultural research. The results section can be improved by adding some analyses based on this guideline as well.

Discussion. The authors thoroughly discuss the findings related to the results section. However, more discussion should be made addressing the matter of cultural generalizations. The current manuscript indeed sounds stereotypical. This can be improved by adding points about potential cultural biases from the current study design and analysis. The small sample sizes collected from college students should be included as limitations. The results should be interpreted with cautions. The data can’t represent the three cultures.

6. PLOS authors have the option to publish the peer review history of their article (what does this mean?). If published, this will include your full peer review and any attached files.

Reviewer #1: **Yes: **Nipat Bock Pichayayothin

---

## [Author Response · Author response to Decision Letter 0]

13 Oct 2022

Dear Reviewer, thank you for your comments. Please see our responses to your comments below. 

Reviewer #1: The current manuscript makes a good contribution to the adult attachment framework. The study investigates the relations of parental bonding and adult attachment style in three different samples from 3 countries. The previous findings of the link between perceived parental bonding and attachment style have been conflicting. The current study attempts to clarify the relationships. Although the current study presents meaningful empirical cross-cultural data, I find that the information presented in the methodology, results, and discussion sections should be revised to improve cross-cultural aspects of the study before accepting for publication. Thus, my decision is to revise and resubmit. My specific feedbacks are as follows.

Abstract. I find the final sentence in this section “However, it does not exclude activation of original Internal Working Models to contexts in which the individual interprets a danger to the relationship” does not best represent many good arguments the authors have in the discussion section. The replacement of this sentence would improve the section.

Thank you for your comment. We have removed the sentence and replaced it with the following one: “These results suggest that different cultural models influence adult attachment representations differently, in terms of the weight placed on attachment-related avoidance, attachment-related anxiety, care, and overprotection in infant and adult relationships.”

Introduction. This section is well-written overall. The review of the previous findings is extensive. The rationale of the study is sound.

Thank you for your comment.

Method and results. These sections need some clarification to strengthen the cross-cultural research approach. The description of measurements used in the study needs additional information about the examination of measurement invariance to ensure the measures used are comparable across cultures. Please see Lacko et al. (2022) https://doi.org/10.1177/10693971211068971 for guideline on statistical analyses to reduce biases in cross-cultural research. The results section can be improved by adding some analyses based on this guideline as well.

Thank you for your comment. We have elaborated on the psychometric properties and factorial structures of the measures used in this study as follows: 

“The PBI-S has been found to demonstrate similar psychometric properties as the PBI and was found to have a two-factor structure, although a three-factor structure appeared to show greater predictive power in relation to affective disorders [121]. The PBI-J used in [123] was demonstrated to show the same two-factor loading in [59] although there have been inconsistent findings of the factorial structure in different Japanese samples [124]. The PBI-I also demonstrated high internal consistency and a two-factor solution was also found [122].” 

“The ECR-S showed the intended two-factor structure in the study conducted by Alonso-Arbiol et al. [115]. There was also evidence indicating high internal consistency, test-retest reliability and criterion and construct validity [115]. The ECR-I was also found to demonstrate the two-factor structure along with adequate internal consistency and test-retest reliability [116]. Finally, the ECR-J also demonstrated the two-factor structure and results indicated adequate internal consistency and construct validity [117].” 

We have also included the analysis of measurement invariance as follows: 

“First, we present the descriptive statistics, Cronbach's alpha, Kolmogorov-Smirnov normality test, and Kruskal Wallis test with Bonferroni Post-Hoc tests for ECR and PBI factors and classifications in the Spanish, Italian, and Japanese groups. To ensure ECR and PBI factors measures are comparable, we ran a multigroup factorial analysis (CFA) to test the measurement invariance and reduce possible biases in cross-cultural research [127]. Four models are explored, establishing a progressive series of restrictions on model parameters to assess the invariance between groups [128, 129]. To assess the differences between the models, we use the variations of the CFI indices (ΔCFI) and RMSEA (ΔRMSEA). The strong invariance is assumed when ΔCFI ≤ 0.01 and ΔRMSEA are ≤ 0.015 [130], while partial invariance is assumed when ΔCFI ≤ 0.01 or ΔRMSEA is ≤ 0.015 [131].”

“To explore the measurement invariance of ECR and PBI factors in the three groups, a multigroup factorial analysis (CFA) was run. The ΔCFI scores (invariance of CFA models) suggest that the ECR (F1. Anxiety; F2. Avoidance) and PBI (F1. Care; F2. Overprotection) bifactorial structures do not fit well, underlying a factor model variance in the three groups (Table 4). On the other hand, the ΔRMSEA (equivalence of metric variance) shows metric invariance in the three groups, suggesting that each item contributed to the latent construct to a similar degree across the Spanish, Italian and Japanese groups (Table 4). In conclusion, the results indicate partial invariance of the ECR metrics in the three groups.”

Table 4 - Multigroup confirmatory factorial analysis (CFA). 

 ① Model X2 (Df) X2/DF ② CFI ③ RMSEA (IC;90%) Comparison ④ ΔX2 ⑤ ΔCFI * ⑥ ΔRMSEA* 

ECR (F1. Anxiety; F2; Avoidance) M1 - Unconstrained (baseline) 4038,56 (1742) 2,27 0,64 0,065 (0,062; 0,067) 

 M2 - Measurement weights 4407,88 (1850) 2,83 0,59 0,068 (0,065;0,070) M2 vs M1 369,32 (108), p < 0,01 -0,05 0,003

 M3- Measurement intercepts 5582,30 (1922) 2,90 0,42 0,079 (0,077;0,082) M3 vs M2 1174,42 (72), p < 0,01 -0,17 0,011

 M4 - Measurement residuals 6079,85 (1988) 3,04 0,36 0,082 (0,080; 0,085) M4 vs M3 497,55 (66), p < 0,01 -0,06 0,003

PBI-M (F1. Care; F2; Overprot.) M1 - Unconstrained (baseline) 1844,46 (825) 2,24 0,73 0,064 (0,060; 0,068) 

 M2 - Measurement weights 1996,61 (871) 2,92 0,71 0,065 (0,062; 0,069) M2 vs M1 152,15 (46), p < 0,01 -0,02 0,001

 M3- Measurement intercepts 2397,89 (921) 2,60 0,62 0,073 (0,069; 0,076) M3 vs M2 431,28 (50), p < 0,01 -0,09 0,008

 M4 - Measurement residuals 2720,42 (975) 2,79 0,55 0,077 (0,074; 0,080) M4 vs M3 322,53 (54), p < 0,01 -0,07 0,004

PBI-F (F1. Care; F2; Overprot.) M1 - Unconstrained (baseline) 1767,81 (825) 2,14 0,77 0,062 (0,058; 0,065) 

 M2 - Measurement weights 1912,63 (871) 2,20 0,75 0,063 (0,059; 0,067) M2 vs M1 144,83 (46), p < 0,01 -0,02 0,001

 M3- Measurement intercepts 2381,30 (921) 2,59 0,65 0,072 (0,069; 0,076) M3 vs M2 468,66 (50), p < 0,01 -0,1 0,009

 M4 - Measurement residuals 2642,263 (975) 2,71 0,60 0,075 (0,072; 0,079) M4 vs M3 260,96 (54), p < 0,01 -0,05 0,003

Values indicating metric invariance are highlighted in grey.

① M1: Unconstrained invariance model (baseline) with free estimation of factor loadings, intercepts, and error variance in each group. M2: Measurement weights model (metric invariance) with equal factor loadings in the groups. M3: Measurement intercepts model (scalar invariance) with equal intercepts and factor loadings in the groups. M4: Measurement residuals model (strict invariance) with equal intercepts, factor loadings, and error variance in the groups.

② CFI - Comparative Fit Index; CFI ≥ 0.95 the model fits the sample [132].

③ RMSEA - Root mean squared error of approximation; RMSEA ≤ 0,05 the model fits the sample [132].

④ ΔX2 Invariance assumed when p> 0,05

⑤ ΔCFI (between-group invariance of CFA models). Factor model invariance assumed when p< 0,01.

⑥ ΔRMSEA (equivalence of metric variance). Metric invariance assumed when p< 0,015.

* Strong invariance is assumed when ΔCFI ≤ 0.01 and ΔRMSEA ≤ 0.015 [130]. Partial invariance is assumed when ΔCFI ≤ 0.01 or ΔRMSEA ≤ 0.015 [131].

Discussion. The authors thoroughly discuss the findings related to the results section. However, more discussion should be made addressing the matter of cultural generalizations. The current manuscript indeed sounds stereotypical. This can be improved by adding points about potential cultural biases from the current study design and analysis. The small sample sizes collected from college students should be included as limitations. The results should be interpreted with cautions. The data can’t represent the three cultures.

Thank you for raising these points. We have added a section in the limitations addressing these points as follows: “Finally, the results in the present study should be interpreted with caution due to the relatively limited sample collected from each country. The participants were also sampled from university students in the three countries, which limits their generalisability to their respective cultures. Although the individualism-collectivism framework and the assumptions of Western cultures as individualistic and Eastern cultures as collectivistic is often applied in cross-cultural research, it is important to note that individualism-collectivism is continuous rather than a dichotomous categorisation and such cultural generalizations may not necessarily reflect each country as a whole. As demonstrated in a number of studies, countries themselves may not be culturally homogenous [138-139]. For example, there are significant regional differences between the Northern and Southern regions of Italy [140] and the sample in the current study was collected from a Northern autonomous province of Italy. Hence, future studies can also look to examine adult attachment within intranational subcultures as well.”

---

## [Decision Letter · Decision Letter 1]

14 Nov 2022

Parental bonding in retrospect and adult attachment style: A comparative study between Spanish, Italian and Japanese cultures

PONE-D-22-15135R1

Dear Dr. Esposito,

We’re pleased to inform you that your manuscript has been judged scientifically suitable for publication and will be formally accepted for publication once it meets all outstanding technical requirements.

Kind regards,

Carla Maria Gomes Marques de Faria, Ph.D.

Academic Editor

PLOS ONE

Additional Editor Comments (optional):

We appreciate the effort made in reviewing the manuscript, respecting the reviewer's criticisms and suggestions. The quality of the manuscript improve substantially, meeting the criteria for publication.

Reviewers' comments:

Reviewer's Responses to Questions

**Comments to the Author**

1. If the authors have adequately addressed your comments raised in a previous round of review and you feel that this manuscript is now acceptable for publication, you may indicate that here to bypass the “Comments to the Author” section, enter your conflict of interest statement in the “Confidential to Editor” section, and submit your "Accept" recommendation.

Reviewer #1: All comments have been addressed

2. Is the manuscript technically sound, and do the data support the conclusions?

Reviewer #1: Yes

3. Has the statistical analysis been performed appropriately and rigorously? 

Reviewer #1: Yes

4. Have the authors made all data underlying the findings in their manuscript fully available?

Reviewer #1: Yes

5. Is the manuscript presented in an intelligible fashion and written in standard English?

Reviewer #1: Yes

6. Review Comments to the Author

Reviewer #1: The authors’ revised manuscript was tremendously improved from the original version. Additional written sections reporting additional cross-cultural measurement analyses and pointing out limitations of the study have appropriately addressed the previous concerns. Thus, I approve this revised version for publication.

7. PLOS authors have the option to publish the peer review history of their article (what does this mean?). If published, this will include your full peer review and any attached files.

Reviewer #1: **Yes: **Nipat Pichayayothin

---

## [Editor Report · Acceptance letter]

15 Nov 2022

PONE-D-22-15135R1 

Parental bonding in retrospect and adult attachment style: A comparative study between Spanish, Italian and Japanese cultures 

Dear Dr. Esposito:

I'm pleased to inform you that your manuscript has been deemed suitable for publication in PLOS ONE. Congratulations! Your manuscript is now with our production department. 

Kind regards, 

on behalf of

Professor Carla Maria Gomes Marques de Faria 

Academic Editor

PLOS ONE